# Effects of *Allium mongolicum* Regel and Its Flavonoids on Constipation

**DOI:** 10.3390/biom10010014

**Published:** 2019-12-20

**Authors:** Yue Chen, Zhijuan Ding, Yuzheng Wu, Qian Chen, Mengyang Liu, Haiyang Yu, Dan Wang, Yi Zhang, Tao Wang

**Affiliations:** 1Institute of Traditional Chinese Medicine, Tianjin University of Traditional Chinese Medicine, 10 Poyanghu Road, Jinghai District, Tianjin 301617, China; yueyue17208@163.com (Y.C.); serafinachen@163.com (Q.C.); nkwangdan@163.com (D.W.); 2Tianjin State Key Laboratory of Modern Chinese Medicine, Tianjin University of Traditional Chinese Medicine, 10 Poyanghu Road, Jinghai District, Tianjin 301617, China; 15222792071@163.com (Z.D.); nxwuyuzheng@163.com (Y.W.); liumengyang0212@tjutcm.edu.cn (M.L.); 3Key Laboratory of Pharmacology of Traditional Chinese Medical Formulae (Tianjin University of Traditional Chinese Medicine), Ministry of Education, 10 Poyanghu Road, Jinghai District, Tianjin 301617, China; hyyu@tjutcm.edu.cn

**Keywords:** constipation, *Allium mongolicum*, AQP3, intestinal motility, G protein alpha, PI3K, calcium flux

## Abstract

Constipation is a common bowel disease in adults with the symptoms of dry stool or difficulty passing stool. Compared with medication therapy, patients show more compliance with the diet therapy, and thus the diet therapy normally exhibits better therapeutic effect. *Allium mongolicum* Regel s a perennial herb of Liliaceae native to Mongolia, Kazakhstan, and China, which is traditionally used for constipation. In this paper, we partly clarify the effectiveness of *A. mongolicum* on constipation from two aspects, including maintaining colon water content and increasing intestinal transit. In loperamide-induced constipation mice model, nine days oral administration of *A. mongolicum* 50% ethanolic extract increased luminal side water content and regulated intestinal movement rhythm to normalize stools. The activity at least partly related to down-regulation of colon aquaporins 3 (AQP3) expression, and up-regulation and activation of G protein alpha (Gα) and phosphoinositide 3-kinases (PI3K). Further, activities on intestine movements were tested using compounds isolated from *A. mongolicum*. Three kinds of major flavonoids significantly increased cellular calcium flux in HCT116 cells and promoted mice intestine smooth muscle contraction. The activity may be related to M choline receptor, μ opioid receptor, 5-HT3 receptor, and inositol 1,4,5-trisphosphate (IP3) receptor.

## 1. Introduction

Constipation is a more common bowel disease in adults. The symptoms of constipation include dry stool or difficulty passing stool [1]. Although constipation is not fatal, long-term abdominal uncomfortable condition lowers life quality of patients. Epidemiological investigation on the global prevalence of constipation revealed its regional characteristics. The highest incidence rates were observed in Oceania (19.7%) and Europe (19.2%), and Asia appears to be relatively low (10.8%) [2].

Except organic changes, generally, constipation was caused by decreased intestinal motility and increased water absorption from the intestine to blood [3]. Intestinal motility was regulated by nervous system and neurotransmitters, such as 5-hydroxytryptamine and acetylcholine. Following the binding of neurotransmitters with their corresponding receptors, voltage-dependent calcium channel was opened and intracellular calcium concentration increased, stimulating the contraction of smooth muscle [4]. Low water content in stool is another key factor of constipation. The water transport was regulated by water channel proteins aquaporins (AQPs). AQP 1–4 and AQP 8 were mainly distributed in colon tissue to facilitate the uptake of water from colon lumenal side to blood. It has been reported that the levels of AQPs are significantly higher in constipation patients than those in the healthy controls [5].

Constipation is a recurrent, lifestyle-related disease. Surrounding to the pathogenesis mechanism, laxative is used to relieve symptom in clinic, including plecanatide and polyethylene glycol (drawing water into colon), correctol and prucalopride (increasing colon motility), etc. Long-term side effects of laxative use include bloating, cramping, diarrhea, nausea, headache, and vomiting [6].

Diet therapy is another method for treating constipation. Several studies have indicated that a high-fiber diet can increase stool excretion and shorten colon transit time, while poor-fiber diet induces constipation [1]. Soluble-fiber-enriched foodstuffs, such as psyllium, are proved to be beneficial for constipation [7]. Not only macro-molecular substances but also low-molecular-weight compounds from edible herb showed laxative effect, such as steroidal saponin spicatoside A (from *Liriope platyphylla*) [8], quercetin [9], and naringenin [10]. In some cases, diet-therapy has better patient compliance than pharmacotherapy, which contributes to better therapeutic effect on constipation.

*Allium mongolicum* Regel is a perennial herb of Liliaceae native to Mongolia, Kazakhstan, and China (Inner Mongolia, Gansu, Ningxia, Xinjiang, et al.) [11]. The chemical constituents of *A. mongolicum* contain isoquercetin, quercetin and its glucosides [12], and kaempferol and its glucosides [13]. The biological activities were reported as antioxidant effects [14], and immuno-enhancement and anti-inflammation [15]. In Inner Mongolia local area, *A. mongolicum* was used to treat gastrointestinal diseases, such as indigestion, constipation, and enteritis [16], but the mechanism remained unknown.

In this paper, the effect of *A. mongolicum* extracts on constipation was investigated by using loperamide-induced rat constipation model. The effects of *A. mongolicum* extracts and its major flavonoids on intestinal motility were studied by isolated mice intestinal tissue to disclose the underlying mechanisms of *A. mongolicum* extracts in the therapy of constipation.

## 2. Materials and Methods

### 2.1. Materials

The fresh aerial parts of *A. mongolicm* were collected from alxa league, Inner Mongolia, and identified by Dr. Tianxiang Li of Tianjin University of Traditional Chinese Medicine (TCM). A voucher specimen (No. 20150905-006) was deposited at the Institute of Traditional Chinese Medicines of Tianjin University of TCM. The fresh aerial parts of *A. mongolicum* (17.8 kg) were cut to segments and immersed in 100 L 50% ethanol under room temperature for 24 h. After filtration, the solvent was removed in vacuo under 40 °C and dried *A. mongolicum* 50% ethanol extract (515.0 g, AM) was obtained.

Loperamide hydrochloride capsule (LOP), nifedipine sustained release tablets (NIF), ondansetron hydrochloride tablets (OND), and Mosapride citrate tablets (MOS) were the products of Xian Janssen Pharmaceutical Ltd. (Xi’an, China), Yangtze River Pharmaceutical Group (Nanjing, China), Qilu Pharmaceutical (Jinan, China), and Jiangsu Hansoh Pharmaceutical Group Co., Ltd. (Lianyungang, China), respectively. 2-Aminoethyl diphenylborinate (2-APB) and ondansetron were obtained from Shanghai Macklin Biochemical Co., Ltd. (Shanghai, China). Acetylcholine (Ach), atropine and dimethyl sulfoxide (DMSO) were purchased from Sigma Chemical Co. (St. Louis, MO, USA). BCA protein assay kit was obtained from Thermo Fisher Sci. Inc. (Waltham, MA, USA). Rabbit anti-β-actin, anti-AQP3, anti-c-kit, anti-SCF, and anti-Gα antibodies were obtained from Abcam plc. (Cambridge, MA, USA). Anti-PI3K and anti-p-PI3K antibodies were purchased from Beijing Biosynthesis Biotechnology Co., Ltd. (Beijing, China). Horseradish peroxidase-conjugated anti-rabbit immunoglobulin G (IgG) was purchased from Beijing Solarbio Science & Technology Co., Ltd. (Beijing, China). Motilin ELISA kit was purchased from Elabscience Biotechnology Co., Ltd. (Wuhan, China).

### 2.2. Animals

ICR mice (male, body weight 22–25 g) were purchased from Vital River Laboratory Animal Technology Co., Ltd. (Beijing, China). All animals were acclimated for 1 week before the experiments, allowed free access to a standard diet and drinking ad libitum, adapted to the experimental conditions at 22 ± 2 °C, and humidity 60 ± 5% with a fixed 12 h artificial light period. All animal experiments were approved by Tianjin University of Traditional Chinese Medicine Committee on Use and Care of Animals (TCM-LAEC20170037).

### 2.3. Loperamide-Induced Mouse Constipation Model

After 1 week adaption, mice were randomly divided into six groups (n = 10), including normal group, control group, positive control group, and AM treated groups (200 and 100 mg/kg). Constipation model was induced by loperamide according to literature reported method with some modifications [17]. Briefly, except normal group, all the animals were orally administrated with LOP saline solution at a final dosage of 3 mg/kg, and the administration volume was 10 mL/kg. The normal group was treated with the same volume of saline. One hour later, AM suspended in saline was orally administrated with the final doses of 100 and 200 mg/kg, and the oral administration volumes were 10 mL/kg body weight. The normal and control groups received saline with the same volume, while the positive control group received MOS saline solution at the final dose of 20 mg/kg. The same treatments were conducted every day for 9 days, and the stool was collected for 2 h on the day before the last administration. Body weights of the mice were recorded every day.

### 2.4. Measurement of Gastrointestinal Transit

On the ninth day, after 30 min of the last administration, 5% carbon powder water suspension was orally administrated to the mice. Thirty minutes later, the mice were sacrificed by cervical dislocation. The small intestine between the pylorus and the cecum was quickly removed. The total length of the small intestine and the distance of carbon powder transmission were measured. The rate of propelling carbon powder in small intestine was calculated based on the distance of carbon powder transmission (CT) and total length of the small intestine (TL) [18], as follows:
Gastrointestinal transit = CT/TL × 100%(1)

### 2.5. Measurement of Serum Motilin Level in Mice

Blood were collected from infraorbital venous plexus of mice, and then allowed to clot for 2 h at room temperature. The samples were centrifuged at 2000× *g* for 10 min at 4 °C, and the supernatants were collected into a new tube. The motilin in serum was measured with mouse motilin ELISA kit according to the instructions. The samples were diluted 40 times for detection. The OD value was measured at 450 nm on a microplate reader (BioTek Winooski, Vermont, USA).

### 2.6. Histopathology of Colon Tissues

One centimeter of the colon tissue near the cecum was fixed with 4% paraformaldehyde for 24 h, then dehydrated with different concentrations of alcohol, and embedded in paraffin. The paraffin sections at 4 µm were stained with hematoxylin and eosin (HE) for conventional morphological evaluation. The microscopic score was calculated as previously reported [19].

### 2.7. RNA Isolation and Real-Time PCR Analysis

The total RNA was isolated from colon tissue by using TRIzol (TransGen Biotech Co., Ltd. Beijing, China) following the manufacturer’s instructions. RNA concentrations were quantified using a NanoDrop2000 (Thermo Fisher Co., Ltd. Waltham, MA, USA). Reverse transcription was carried out according to the instructions of High-Capacity cDNA Reverse Transcription Kits (Applied Biosystems, Waltham, MA, USA). The cDNA was stored in a refrigerator at −80 °C for further use. The primers used for real-time PCR were synthesized by Dingguo Bio Co. Ltd., (Shanghai, China), and the sequences were listed in Table 1. Real-time PCR analysis was performed with the SYBR Green Quantity Tech RT-PCR kit (Applied Biosystems, Waltham, MA, USA) on ABI 7500 System (Applied Biosystems, Woodlands, Singapore). The expressions of all target genes were normalized with β-actin following the 2^−ΔΔCt^ methods.

### 2.8. Western Blot Analysis

Protein sample of colon tissue was separated by SDS-PAGE electrophoresis, and then transferred to a polyvinylidene difluoride membrane (Millipore, Bedford, MA, USA). The membrane was blocked by 5% skim milk for 1 h. After an overnight incubation of primary antibody at 4 °C, the membrane was washed three times in Tris-buffered saline with Tween 20 (TBST, Beijing Solarbio Science & Technology Co. Ltd., Beijing, China). The secondary antibody was incubated with horseradish peroxidase labeling at room temperature for 1 h. Membranes were washed three times with TBST. The expressions of the target proteins were analyzed with ECL method using a chemiluminescence detection kit (Millipore Co. Ltd. MA, USA). The transferred proteins were visualized with ChemiDoc MP Imaging System (Bio-Rad, Hercules, CA, USA). ImageJ 1.50i (NIH, Bethesda, Maryland, USA) (Java 1.60_20) (http//imagej.nih.gov/ij) was used to calculate the intensity of western blot. The quantification measurements of western blot results were performed according to literature reports [20].

### 2.9. Isolated Intestinal Tissue Preparation and Isometric Measurements

Mice intestinal tissue (containing jejunum and ileum) preparation and isometric measurements were performed as described previously [21]. Shortly, mice were sacrificed by cervical dislocation after fasting 24 h. The intestinal tissue section was isolated and approximately 1 cm section was cut down. The section was longitudinally connected at the tension converter (Johnson & Johnson Medical China Ltd., Beijing, China), and the tissue was placed in 10 mL Tyrode buffer (1 L contains NaCl 8.0 g, CaCl_2_ 0.2 g, KCl 0.2 g, MgCl_2_ 0.1 g, NaHCO_3_ 1.0 g, KH_2_PO_4_ 0.05 g, and glucose 1.0 g; pH 7.4) at 37.0 °C bubbling with 95% O_2_ and 5% CO_2_ gas. Intestine contractions were recorded using the Power Lab system and the Chart 7 software (AD instrument Ltd., New South Wales, Australia). The isolated intestinal tissue was orderly exposed to atropine (5 ng/mL), OND (5 µg/mL), LOP (12.5 ng/mL), NIF (5 ng/mL), and 2-APB (5 µmol/L) for mechanism study.

### 2.10. Effects of AM and its Compounds on Cellular Calcium Flux in HCT 116 Cells

HCT116 cells were obtained from the cell center at the Chinese Academy of Medical Science and Peking Union Medical College (Beijing, China), cultured in Dulbecco’s modified Eagle medium (DMEM) medium supplemented with 10% fetal bovine serum (FBS) and 100 µg/mL penicillin/streptomycin.

HCT116 cells were seeded at a density of 3 × 10^5^ cells/mL in a 96-well plate with a clear black bottom. After 24 h of incubation, the medium was removed, the cells were washed three times with HEPES buffer, and each well was filled with 70 µL Fluo-4 acetyl-D-methionine (2 µmol/L in DMSO, Solarbio Science & Technology Co., Ltd., Beijing, China) as working solution. After incubation for 1 h in the dark at 37 °C, the Fluo4-AM solution was discarded. The cells were carefully washed three times with HEPES solution, and then incubated for another 30 min in the dark at 37 °C, while 200 µL HEPESs buffers, Ach, and the AM were added. At the meantime, calcium flow changes of the cells were continuously detected by a multifunctional microplate reader (BioTek Winooski, Vermont, USA) at a 10 s interval for 5 min [22]. Finally, the flavonoids of AM, various inhibitors (OND, 2-APB, atropine, LOP, NIF), and combination of inhibitors and flavonoids of AM were added to the plate, and calcium flux was measured in the same way.

### 2.11. Statistical Analysis

The statistical analysis was performed with SPSS 11.5 (IBM, Armonk, New York, USA) Significant differences between groups were evaluated by one-way ANOVA, and Tukey’s Studentized range test was used for post hoc evaluations. The data were reported as mean ± S.E.M. The differences were considered significant when *p* < 0.05.

## 3. Results

### 3.1. AM Alleviate Symptoms of LOP-Induced Constipation in Mice

The improvement of constipation was primarily assessed by increased defecation and enhanced intestinal movement. The effect of AM was evaluated by a LOP-induced constipation mouse model. Figure 1A showed that the body weights of the mice were not changed in either the LOP-induced constipation mouse model group or the treated group. Compared with the normal group, the fecal excretion in the control group was significantly decreased by 48.7% (Figure 1B, *p* < 0.001). Compared with the control group, treatment with 200 or 100 mg/kg AM significantly increased fecal excretion by 96.7% (Figure 1B, *p* < 0.01) and 46.3% (Figure 1B, *p* < 0.05), respectively. This trend of increasing defecation is consistent with the MOS-administrated group, which was settled as a positive control group.

MTL is a hormone expressed along the gastrointestinal tract, which can promote the movement of the intestinal tract [23]. As it is shown in Figure 1C, LOP administration resulted in a significant reduction of MTL concentration in serum by 28.8% (*p* < 0.05) compared with the normal group. However, when treated with 200 mg/kg AM, the serum MTL concentration was significantly increased by 16.4% (*p* < 0.01) compared with the control group. A similar trend was observed in positive control group. The gastrointestinal motility was assessed by carbon powder transit experiment. In this study, the gastrointestinal transit was decreased by 19.8% (Figure 1D, *p* < 0.01) in the control group as compared with the normal group. Compared with the control group, administration of 200 or 100 mg/kg AM significantly increased gastrointestinal transit by 43.7% (Figure 1D, *p* < 0.001) and 34.2% (Figure 1D, *p* < 0.05), respectively. These results suggested that AM could improve the symptoms of LOP-induced constipation in mice.

### 3.2. AM Improved Pathological Symptoms of LOP-Induced Constipation in Mice

Goblet cells in intestine are responsible for producing mucins, such as MUC2, which play vital roles in lubrication of the intestinal tract [24]. HE staining showed that the number of colonic goblet cells in control group was decreased than that in the normal group. However, treatment with MOS or AM made the number of goblet cells to be in the normal range (Figure 1E). Furthermore, the thickness of colonic wall in LOP-induced mouse control group was thinner than that in the normal group. Nevertheless, the thickness of colonic wall becomes normal after MOS or AM therapy These results indicated that AM could maintain the number and form of intestinal goblet cell at a normal level.

### 3.3. AM Regulate the Expressions of AQP3 and Intestinal-Motility-Related Protein Expression in LOP-Induced Constipation in Mice

Aquaporins 3 (AQP3) is a water channel protein, which is predominately distributed in the gastrointestinal tract to keep the water content of the feces [25]. It has been reported that the increase of AQP3 in colon is associated with the development of constipation in rats [26]. Therefore, in this study, the expression of AQP3 in the colon of mice was investigated. After the administration of LOP for 9 days, mRNA level of AQP3 was significantly increased in control group compared with the normal group (Figure 2A, *p* < 0.05). However, when compared with the control group, the expression of AQP3 was significantly decreased by 79.1% (Figure 2A, *p* < 0.001) and 62.6% (Figure 2A, *p* < 0.001) after treatment with 200 or 100 mg/kg AM, respectively. Moreover, a decrease of AQP3 mRNA level was also observed in positive control group (Figure 2A, *p* < 0.05). In agreement with the mRNA expressions, western blot analysis showed that the protein level of AQP3 was significantly increased in the colon of LOP-induced constipation mice by 30.9% (Figure 2B,C, *p* < 0.001) compared with the normal group. After treatment with AM at the dosage of 200 or 100 mg/kg, the protein level of AQP3 was significantly decreased by 47.4% (Figure 2B,C, *p* < 0.001) and 44.5% (Figure 2B, *p* < 0.001), respectively, when compared with control group.

To investigate whether AM could affect the genes responsible for regulation of intestinal motility, we measured the mRNA expressions of Gα, c-kit, PI3K and SCF. It has been reported that negative modulation of the voltage-dependent L-type Ca^2+^ channel (LTCC) by activated Gα submit in cardiac myocytes is mediated by inhibition of PI3K [27]. LOP administration (control group) enhanced the mRNA level of Gα by 32.4% (Figure 2F, *p* < 0.05) and 24.6% (Figure 2F, *p* < 0.05) compared with the normal group and positive control group, respectively. On the contrary, the mRNA level of Gα were suppressed by 32.4% (Figure 2F, *p* < 0.01) and 31.7% (Figure 2F, *p* < 0.05) after AM administration at the dosage of 200 or 100 mg/kg, respectively. The mRNA levels of c-kit, PI3K, and SCF in control group were significantly decreased when compared with normal group by 56.2% (Figure 2D, *p* < 0.05), 43.3% (Figure 2E, *p* < 0.05), and 97.8% (Figure 2G, *p* < 0.05), respectively. Yet the mRNA levels of c-kit in AM-treated groups were higher than control group by 139.1% (Figure 2D, *p* < 0.05) and 59.0% (Figure 2D), respectively. Moreover, the mRNA expression of PI3K under AM therapy at the dosage of 200 or 100 mg/kg were increased by 69.6% (Figure 2E, *p* < 0.05) and 33.9% (Figure 2E, *p* < 0.05), respectively, when compared with the control group. And the mRNA level of SCF was dramatically increased by 783.7% (Figure 2G, *p* < 0.05) and 204.7% compared with the control group, after AM therapy at the dosage of 200 or 100 mg/kg, respectively. As shown in Figure 2H–J, the protein expressions of p-PI3K, c-kit, and SCF were remarkably down-regulated, and the protein expressions of PI3K were increased in control group compared to normal group that were restored in the AM-treated group. Taken together, these data indicate a protective role of AM in regulating the intestinal-motility-related genes and proteins in LOP-induced constipation mice.

### 3.4. AM and its Flavonoids-Promoted Mice Intestine Smooth Muscle Contraction

Isolated intestine was used to assess the effects of AM on intestinal smooth muscle spontaneous contraction. As shown in Figure 3A–D, AM (200 µg/mL) dramatically promoted the frequency and the active amplitude on intestine smooth muscle compared to the normal group. The MOS (200 µg/mL)-treated group increased active amplitude comparing with normal group, while active tension was not altered by the AM extracts or MOS treatments (Figure 3D).

As in previous report, caffeic acid type glycoside (mongophenoside A1, **1**), kaempferol type glycoside (kaempferol-3-*O*-*β*-D-rutinoside, **2**), and quercetin type glycoside (quercetin-3,4′-di-*O*-*β*-D-glucopyranoside, **3**) were major compounds in AM (Figure 4A) [12,13,14]. Consequently, their activities on active amplitude from AM were tested using the same system. Treatment with compounds **1**–**3** significantly increased the active amplitude at 50 µM (Figure 4B,D).

Furthermore, several kinds of antagonists were used to study the possible mechanism of **1**–**3** (Figure 4A and Appendix A). Atropine (5 ng/mL, M choline receptor antagonist), loperamide (12.5 ng/mL, μ opioid receptor antagonist), ondansetron (5 µg/mL, 5-HT3 receptor antagonist), 2-APB (5 µM, IP3 receptor antagonist), and nifedipine (5 ng/mL, calcium channel blocker) were used. Atropine, 2-APB, and NIF could completely antagonise the activation on amplitude of compound **1**–**3**, respectively. LOP could entirely offset the excitatory effect of **1** and **2** (Figure 5B,G), but not reverse the effect of **3** (Figure 5L). Then, OND could entirely inhibit the effect of **2** and **3** (Figure 5J,O), but not suppress the effect of **1** (Figure 5E).

### 3.5. Compounds of AM-Stimulated Calcium Influx

The intracellular calcium concentration was a crucial factor to trigger contraction of the small intestine. A HCT116 cell line with fluorescent probe was used to test calcium influx. The recorded Ca^2+^ sensitive curves were shown in Appendix A. As shown in Figure 6, **1**–**3** significantly enhance the calcium influx at the dosage of 50 µM compared with the normal group. Atropine, 2-APB and NIF could completely antagonise the increased calcium influx of **1**–**3** on HCT116 cells, respectively. LOP could entirely offset the activation effect of **1** and **2** (Figure 6B,G), but not reverse the effect of **3** (Figure 6L). And then, OND could entirely inhibit the effect of **2** and **3** (Figure 6J,O), but not suppress the effect of 1 (Figure 6E). These results were agreed with promotion effects of compounds **1**–**3** on intestine smooth muscle contraction. It was suggested that the effects of **1**–**3**, at least, may be related to M choline receptor, IP3 receptor, and calcium channel, respectively. Differing from **2**, **1** showed no relation with 5-HT3 receptor and **3** showed no relation with μ opioid receptor

## 4. Discussion

Constipation is a common chronic gastrointestinal disorder. It can be divided into functional type and organic type [28]. Functional constipation is defined as lumpy or hard stools and infrequent bowel movements without organic or structural diseases [29]. The cause of functional constipation still unclear; mental stress, diet digestive or nutritional problem are thought to be the major reasons [30]. Although there are some medications used to treat constipation, long-term abuse of cathartic medication or specific tools have a negative impact on quality of life, and even can induce bowel pathological changes as colon melanosis [31]. Compared with medication therapy, patients with constipation show more compliance with the diet therapy, and thus the diet therapy normally exhibits better therapeutic effect [32]. *A. mongolicum* is a common foodstuff in desert area in north of China, which is considered useful for improving constipation, but the scientific evidence is rarely reported. In this paper, we tried to clarify the effectiveness of *A. mongolicum* from two aspects, including maintaining colon water content and increasing intestinal transit.

Water content in colon is a key factor in development of constipation. AQPs, the water and glycerol selective transporters, ultimately control water absorption from the luminal side to the vascular side [24]. In diarrhea-type ulcerative colitis patient, AQP3 level in colon was significantly reduced compared with healthy volunteers [33], while it was increased in morphine-treated constipated animals [34]. These results indicated that AQP3 plays an important role in the absorption of water in colon. Accompanied by reduction of AQP3 expression, long-term functional constipations showed significant pathological features in colon with a thin colon wall and a decreased colonic goblet cell number [35]. AM significantly promoted the fecal excretion, increased the water content of feces, and improved the thickness of colon wall and the number of goblet cells in the gland. The protein expression analysis showed that AM up-regulated the expression of AQP3 in LOP-induced constipation mice, which indicated that AM may increase luminal side water content to improve constipation.

Abnormal contraction rhythm leads to slow intestinal transit, which has been proved to have a critical role in the pathogenesis of constipation [36]. Interstitial cell of Cajal (ICC) are pacemaker cells in the small intestine. C-kit, a transmembrane protein, is a marker of ICC. SCF, a ligand for c-kit, promotes growth and differentiation of the ICC, and maintains its normal physiological function [37]. AM significantly up-regulated the expression levels of c-kit and SCF, which may be useful for regulating ICC cells, and improving intestinal movement rhythm [37].

Physiologically, contraction of intestinal smooth muscle depends on increasing the intracellular calcium ions, following a subsequent Ca^2+^-triggered actin–myosin cross-bridge formation [38]. The changes of intracellular Ca^2+^ concentration is regulated by Gα protein–mediated IP3 signaling pathway [39,40]. After activation of G-protein coupled receptors (such as Gα) and PI3K, plasma membrane lipid phosphatidylinositol-4,5-bisphosphate (PIP2) is converted to phosphatidylinositol-3,4,5-trisphosphate (PIP3), and increases IP3 levels in the cytosol. Subsequently, IP3 bound to IP3 receptors (IP3Rs) in the membrane of the endoplasmic reticulum (ER) leads to Ca^2+^ release from ER to specific target sites [41,42,43]. AM significantly increased the length of carbon powder indicating the improvement effect on intestinal transit, which may be related to up-regulation and activation of Gα and PI3K.

Oxidative stress plays an important role in constipation [44], including down-regulation of cuprozinc-superoxide dismutase, manganese superoxide dismutase, and catalase, and up-regulation of nitric oxide synthase and its product NO. Literature reported that AM had antioxidant and anti-inflammatory activities [14,15], which may be a possible mechanism of AM on constipation.

To further clarify the mechanism of AM on intestinal motility, the compounds from AM was tested using cell calcium influx and isolated intestine. As our previous reports [12,13], there are three series of compounds in AM, which were caffeic acid type glycoside, kaempferol type glycoside and quercetin type glycoside. Compounds **1**–**3** were selected as typical compound of AM to clarify the possible activity mechanism on constipation. These compounds improved the calcium influx from extracellular to intracellular and increased the contraction amplitude of isolated intestine [45,46], which may be related to M choline receptor, μ opioid receptor, 5-HT3 receptor, IP3 receptor, and calcium channel blocker.

## 5. Conclusions

As a summary, in this paper, we firstly reported the improvising effect of *A. mongolicum* on constipation, and partly revealed the active substances on intestinal motility. This study indicated that *A. mongolicum* is beneficial to manage functional constipation.

## Figures and Tables

**Figure 1 biomolecules-10-00014-f001:**
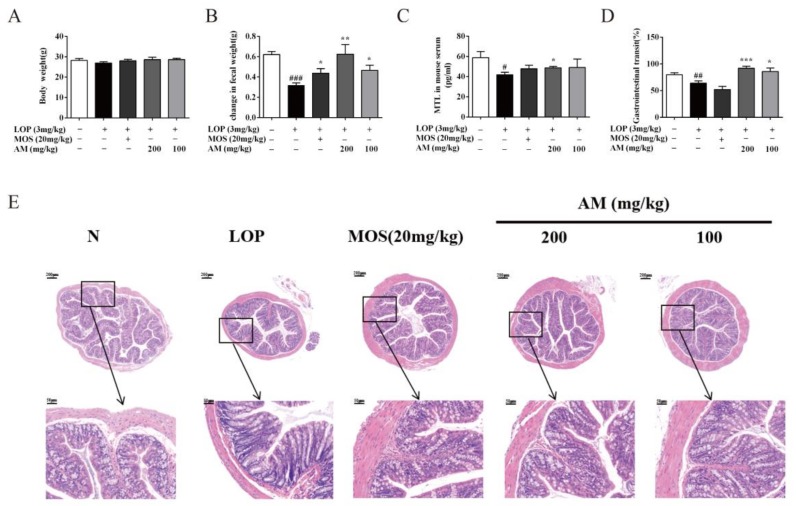
AM alleviate symptoms in loperamide hydrochloride (LOP)-induced constipation mice. (**A**) Body weight; (**B**) change of fecal weight; (**C**) MTL in mouse serum; (**D**) gastrointestinal transit, and (**E**) representative image of hematoxylin and eosin (H&E) staining (n = 10). Magnification shown is 5 µ and scale bar represents 200 µm. The area within the rectangle in each picture is enlarged and presented below; correspondingly, magnification shown is 10 µ and scale bar represents 50 µm. N, normal group; LOP, control group; MOS (mosapride citrate), positive control group; AM 200, AM 200 mg/kg + LOP group; AM 100, AM 100 mg/kg + LOP group. Data are presented as mean ± SEM, # *p* < 0.05, ## *p* < 0.01, and ### *p* < 0.001 versus normal group and *, **, and *** indicate *p* < 0.05, *p* < 0.01and *p* < 0.001 versus control group, respectively.

**Figure 2 biomolecules-10-00014-f002:**
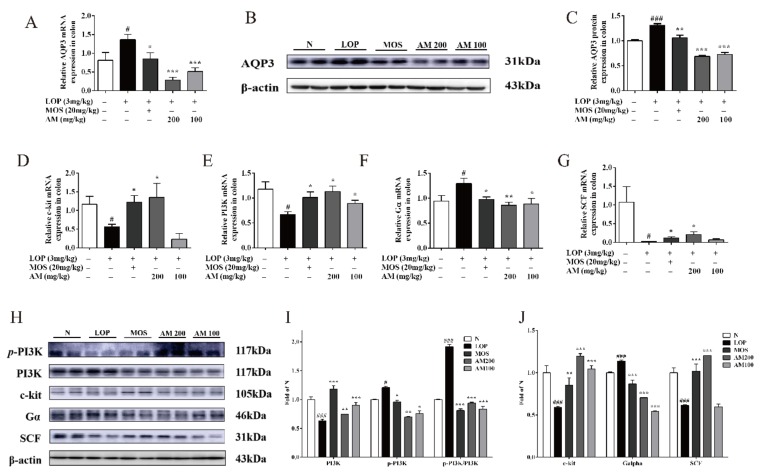
Effect of AM ameliorated intestinal motility related genes and proteins expression in LOP-induced constipation. (**A**–**C**) Relative mRNA and protein level of aquaporins 3 (AQP3) were increased by RT-PCR and western blot in colon tissue of LOP-induced constipation mice (n = 10). (**D**–**G**) RT-PCR analysis for c-kit, PI3K, and Gα and SCF in colon tissue of LOP-induced constipation mice (n = 10). (**H**–**J**) Relative protein level of p-PI3K, PI3K, c-kit, and Gα and SCF were detected in colon tissue of LOP-induced constipation mice (n = 10). N, normal group; LOP, control group; MOS, positive control group; AM 200, AM 200 mg/kg + LOP group; AM 100, AM 100 mg/kg + LOP group. Data are presented as mean ± SEM, # *p* < 0.05, ## *p* < 0.01, and ### *p* < 0.001 versus normal group and *, **, and *** indicate *p* < 0.05, *p* < 0.01, and *p* < 0.001 versus control group, respectively.

**Figure 3 biomolecules-10-00014-f003:**
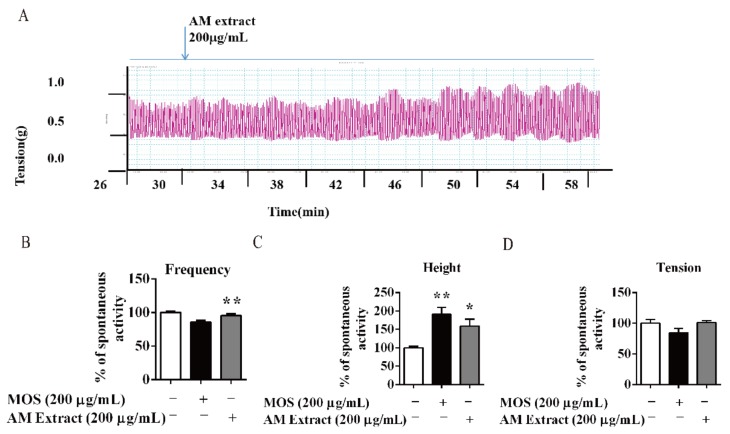
AM promoted intestine smooth muscle movement of isolated intestine of mice. (**A**) The isolated intestine strips were longitudinally mounted in an organ bath, and the contractile response to AM was recorded. (**B**–**D**) Active frequency, height, tension, and spontaneous contractions of isolated intestine smooth muscle. Data are expressed as means ± SEM (n = 6); * and ** indicate *p* < 0.05 and *p* < 0.01, respectively.

**Figure 4 biomolecules-10-00014-f004:**
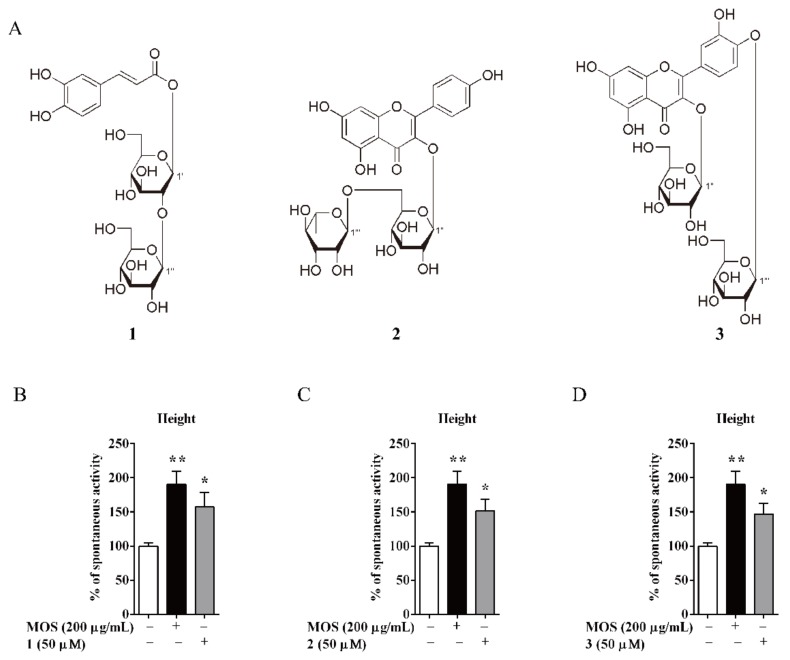
The structures of compounds in AM. (**A**) **1**, mongophenoside A1; **2**, kaempferol-3-O-β-D-rutinoside; **3**, quercetin-3,4′-di-O-β-D-glucopyranoside. (**B**–**D**) Active amplitude of isolated intestine smooth muscle. Data are expressed as means ± SEM (n = 6); * and ** indicate *p* < 0.05 and *p* < 0.01, respectively.

**Figure 5 biomolecules-10-00014-f005:**
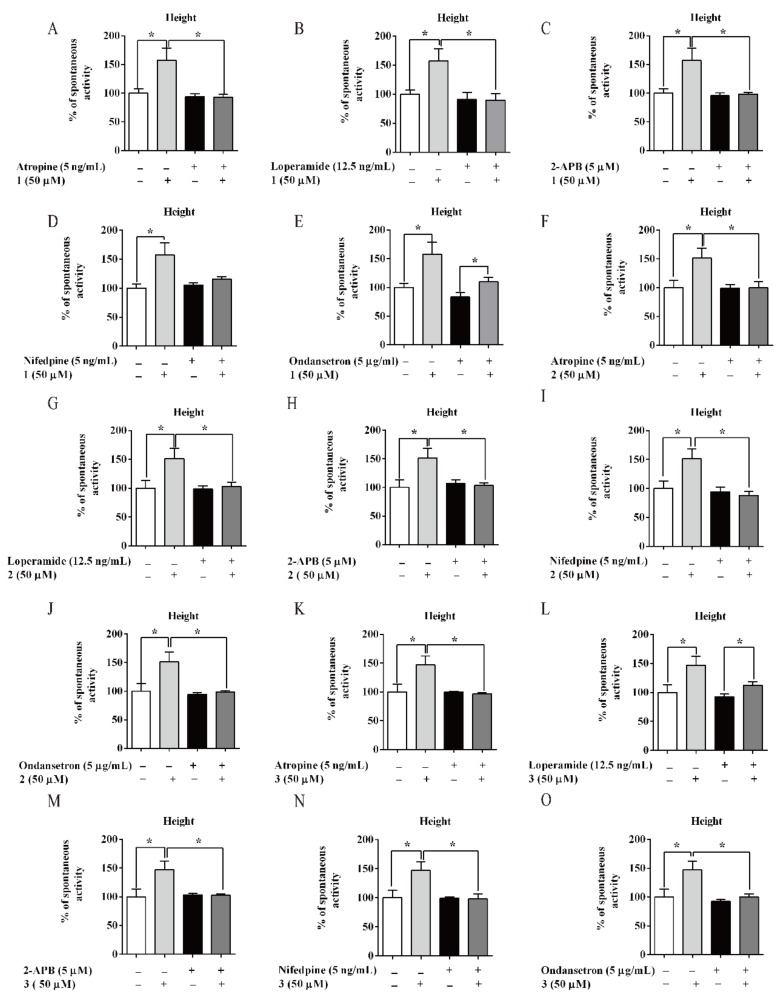
Promotion mechanism of AM flavonoids on mice isolated intestine. Effects of (**A**–**E**) **1**, (**F**–**J**) **2,** and (**K**–**O**) **3** on isolated intestinal contraction with different inhibitors. Data are expressed as means ± SEM (n = 6); * indicate *p* < 0.05.

**Figure 6 biomolecules-10-00014-f006:**
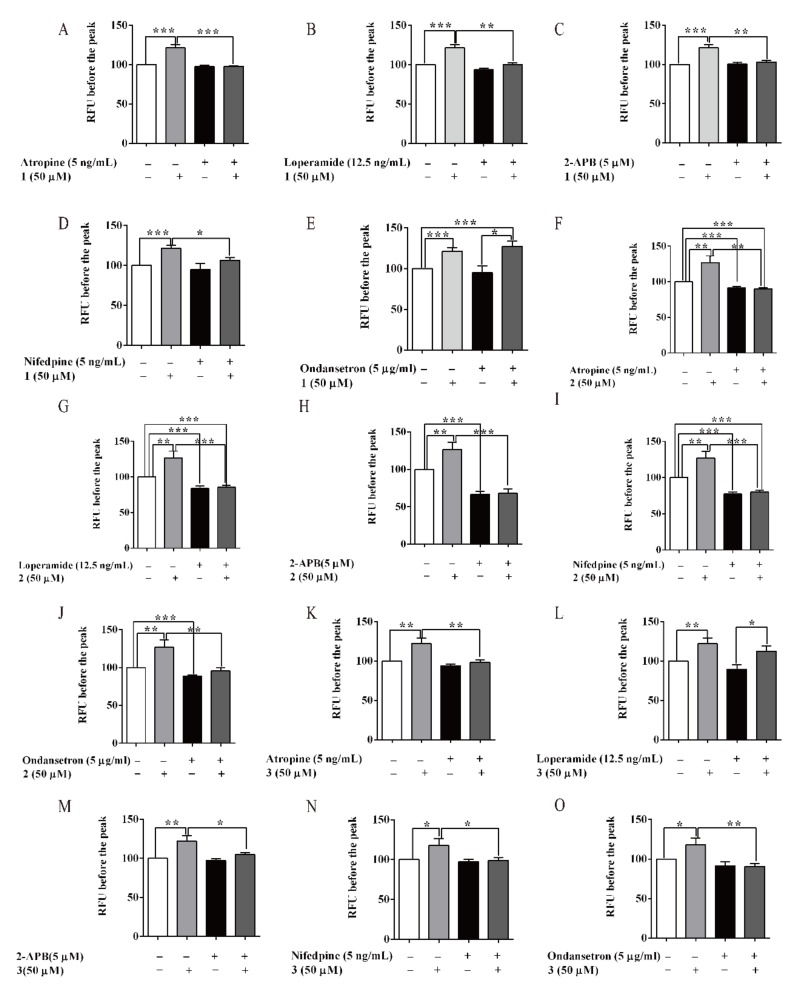
Flavonoids of AM stimulated calcium influx in HCT 116 cells. (**A**–**O**) Effect of (A–E) **1**, (**F**–**J**) **2** and (**K**–**O**) **3** on calcium influx with different inhibitors. Data are expressed as means ± SEM (n = 6); *, **, and *** indicate *p* < 0.05, *p* < 0.01, and *p* < 0.01 respectively.

**Table 1 biomolecules-10-00014-t001:** Sequences of the primers for real-time RT-PCR analysis.

Gene	Sequence
*AQP3*(mouse)	Forward: 5′- GCCAAGGTAGGATAGCAAATAA -3′
	Reverse: 5′- TTGAAAACTTGGTCCCTTGC -3′
*c-kit*(mouse)	Forward: 5′- CCGACGCAACTTCCTTATGAT -3′
	Reverse: 5′- TCAGGACCTTCAGTTCCGACA -3′
*SCF*(mouse)	Forward: 5′- ATAGTGGATGACCTCGTGTTA -3′
	Reverse: 5′- GAATCTTTCTCGGGACCTAAT -3′
*PI3K*(mouse)	Forward: 5′- TGTGTTCTCTGCTCGTCAGG -3′
	Reverse: 5′- TTCTGTAGTGTGGGGGTCCA -3′
*Gα*(mouse)	Forward: 5′- CCCCAGCAGGTTCCTAAGAC -3′
	Reverse: 5′- CGGTCAGGCAAGTAGGAAGG -3′
*β-actin*(mouse)	Forward: 5′- CTGTGCCCATCTACGAGGGCTAT -3′
	Reverse: 5′- TTTGATGTCACGCACGATTTCC -3′

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
