# Peer review of "Effects of *Allium mongolicum* Regel and Its Flavonoids on Constipation"

_biomolecules, 2019, doi:10.3390/biom10010014_

Round 1
Reviewer 1 Report
A generally well-written and informative manuscript. Numerous minor English corrections need to be cleaned up by editorial staff. The manuscript would have been better if the flavonoid data had been omitted. The jump to assuming that the biological effects are due to the flavonoids is logical, but not proven in this study. There should have been a more complete discussion of the secondary metabolites in the allium extract, and the flavonoid contents and amounts. Experiments with the flavonoids should have been left for a separate study. Many vegetables have very similar compounds, so based on the flavonoid argument, most vegetables and fruits should have similar effects on constipation.
The research in this study used an AM extract prepared by a the Chemistry and Analysis Laboratory at Tianjin University of Traditional Chinese medicine. But there is NO description of the age and condition of the plant, the portion of the plant extracted with ethanol, nor the extraction method. Most importantly, there is no mention of the extract concentration, and so this research cannot be reproduced. This is a major flaw.
The concentrations used in the experiments shown in Figures 5 and 6 are missing. The results cannot be fully understood.
The scope of the targets pertinent to intestinal function was extensive and provided quite a complete analysis of the potential biological effects of AM extract. But, what was in the AM extract, and was there any effect of the ethanol?
I am not intimately familiar with this topic, but the methods used, and the tests that were selected appeared appropriate. The combination of the in vivo mice studies with the invitro cell studies made a complete investigation. It is interesting that there is no dose dependency in the AM extract amount on the AQP3 protein levels, but there's a clear dose dependency for AM on increased fecal excretion. There are obviously several layers of control, and some of this is brought out in lines 247-277. Make sure that this is made clear in the Discussion.
Author Response
Response
Reviewer #1:
A generally well-written and informative manuscript.
Q1. Numerous minor English corrections need to be cleaned up by editorial staff.
A1. Thank you for your kind advise. We invited Dr. Rutao Cui (The Laboratory of Melanoma Research in the Section of Cancer Pharmacology, Boston University School of Medicine, E-mail: rutaocui@bu.edu) to revised this manuscript. The revised parts were marked by Track Changes model.
Q2. The manuscript would have been better if the flavonoid data had been omitted. The jump to assuming that the biological effects are due to the flavonoids is logical, but not proven in this study. There should have been a more complete discussion of the secondary metabolites in the allium extract, and the flavonoid contents and amounts. Experiments with the flavonoids should have been left for a separate study. Many vegetables have very similar compounds, so based on the flavonoid argument, most vegetables and fruits should have similar effects on constipation.
A2. Thank you for your kind advise. The proposal of this research is to find the function of AM, and clarify the possible active compound. This is beneficial to further functional food or supplement developments and their quality control. So, we tested the activity of AM extract, and then carried out the phytochemical research to identify the major constituents. The phytochemistry research were published as literatures 12 and 13. The results indicated that there are 3 kinds series compounds in AM, which were caffeic acid type glycoside, kaempferol type glycoside and quercetin type glycoside. Hence, we selected compounds 1-3 as typical compound of AM extract to clarify the possible activity mechanism on constipation. We are very sorry that we cannot omit flavonoid data, because it is very important in this paper.
According to your suggestion, we pended the corresponding constituents of AM in the part of "Discussion". (line 427-430)
We agree with your opinion, these compounds were widely distributed in vegetables and fruits. Without corresponding study, we cannot confirm whether the other vegetables and fruits had the same activity.
Q3. The research in this study used an AM extract prepared by a the Chemistry and Analysis Laboratory at Tianjin University of Traditional Chinese medicine. But there is NO description of the age and condition of the plant, the portion of the plant extracted with ethanol, nor the extraction method. Most importantly, there is no mention of the extract concentration, and so this research cannot be reproduced. This is a major flaw.
A3. Thank you for your comments. We added the details of AM and its extraction method in “line 81-83”. “The fresh aerial parts of AM (17.8 kg) were cut to segments and immersed in 100 L 50% ethanol under room temperature for 24 h. After filtration, the solvent was removed in vacuo under 40°C and dried extract (515.0 g) was obtained.”
Q4. The concentrations used in the experiments shown in Figures 5 and 6 are missing. The results cannot be fully understood.
A4. We are very sorry for our negligence of the concentrations used in the experiments shown in Figures 5 and 6. We have made corrections according to the Reviewer’s comments in the parts “Figures 5 and Figure 6”.
Q5. The scope of the targets pertinent to intestinal function was extensive and provided quite a complete analysis of the potential biological effects of AM extract. But, what was in the AM extract, and was there any effect of the ethanol?
A5. The major components of AM extract are caffeic acid type glycosides, kaempferol type glycosides and quercetin type glycosides as described in “line 427-430”. The procedure of AM extract preparation has been illustrated in the amended manuscript (line78-83). AM was extracted with 50% ethanol, then the extraction liquid was removed completely. There was no ethanol in the final extract.
Q6. I am not intimately familiar with this topic, but the methods used, and the tests that were selected appeared appropriate. The combination of the in vivo mice studies with the invitro cell studies made a complete investigation. It is interesting that there is no dose dependency in the AM extract amount on the AQP3 protein levels, but there's a clear dose dependency for AM on increased fecal excretion.
A6. As it was shown in Figure 2, although there is no clear dose dependency on AQP3 protein levels, there is a clear dose dependency of mRNA level of AQP3. The quantitation of target genes expression was obtained by 2-ΔΔCt methods. The protein quantitation was obtained by gray value analysis method, of which system error is larger than gene expression. We speculated that this is the major reason of no dose dependency of AQP3 protein levels.
On the other hand, improvement of fecal excretion may not only due to the increase of water content in stool, but also due to other factors, such as enhancement of intestinal contraction. Therefore, the dose dependency effects of AM extract on defecation may not only related with the change of AQP3 expression.
Q7. There are obviously several layers of control, and some of this is brought out in lines 247-277. Make sure that this is made clear in the Discussion.
A7. Thank you for the suggestion, we revised throughout the manuscript and specified the mice without any treatment as “normal group”, the mice treated with LOP as “control group”, and the mice treated with MOS as “positive control group”.
Reviewer 2 Report
The article by Yue Chen et al., Some suggestions:
1. Who identified Allium mongolicum Regel and has the voucher specimen.
2. Please justify why I use mouse and not rats in your experiments.
3. In figure 2 it can be more specific to what each bar represents. thanks
4. Do I use a computer program to analyze and present the Western blot?
5. You can expand the discussion on the mechanisms of action of Allium mongolicum Regel. Will it have an antioxidant and anti-inflammatory effect?
6. Which of the three components of Allium mongolicum Regel has the beneficial effect to manage functional constipation
Author Response
Q1. Who identified Allium mongolicum Regel and has the voucher specimen.
A1. Thank you for your kind advice, we revised the corresponding part at “2.1 Materials, line 78-81” as follows: “The fresh aerial parts of AM were collected from alxa league, Inner Mongolia, and identified by Dr. Tianxiang Li of Tianjin University of Traditional Chinese Medicine (TCM). A voucher specimen (No. 20150905-006) was deposited at the Institute of Traditional Chinese Medicines of Tianjin University of TCM.
Q2. Please justify why I use mouse and not rats in your experiments.
A2. According to literature report, there were no significant differences between rats and mice in the study of intestinal movements. Both mice and rats were used to set up constipation animal models in different studies.
Reference that use mice to set up constipation animal models:
[1] Lactobacillus plantarum CQPC02-Fermented Soybean Milk Improves Loperamide-Induced Constipation in Mice. J Med Food. 2019, 1–14. DOI: 10.1089/jmf.2019.4467
[2] Laxative Effect of Spicatoside A by Cholinergic Regulation of Enteric Nerve in Loperamide-Induced Constipation: ICR Mice Model. Molecules. 2019, 896. DOI: 10.3390/molecules24050896
Reference that use rat to set up constipation animal models:
[1] Aqueous Extracts of Herba Cistanche Promoted Intestinal Motility in Loperamide-Induced Constipation Rats by Ameliorating the Interstitial Cells of Cajal. Evidence-Based Complementary and Alternative Medicine. 2017,13. DOI: 10.1155/2017/6236904
[2] Quercetin promotes gastrointestinal motility and mucin secretion in loperamide-induced constipation of SD rats through regulation of the mAChRs downstream signal. PHARMACEUTICAL BIOLOGY. 2018,309-317. DOI: 10.1080/13880209.2018.1474932
In addition, mice require less sample compared to rats, which avoid waste of resources, so we selected mice in this study.
Q3. In figure 2 it can be more specific to what each bar represents. Thanks
A3. Thank you for the suggestion, we have made some adjustments in “Figure 2”. The legend for bar has been added.
Q4. Do I use a computer program to analyze and present the Western blot?
A4. As your suggestion, we revised the methods used to analyze and present the Western blot in the part of method (line 164-167). The transferred proteins were visualized with ChemiDoc MP Imaging System (Bio-Rad, Hercules, CA, USA). ImageJ software was used to calculate the intensity of western blot. The quantification measurements of western blot results were performed according to the literature reports.
Q5. You can expand the discussion on the mechanisms of action of Allium mongolicum Regel. Will it have an antioxidant and anti-inflammatory effect?
A5. Thank you for the kind advice, we have revised the discussion regarding the mechanisms of action of Allium mongolicum Regel as follow: “ Oxidative stress plays important role on constipation, including down-regulation of cuprozinc-superoxide dismutase, manganese superoxide dismutase and catalase, up-regulation of nitric oxide synthase and its product NO. Literature reported that AM extract had antioxidant and anti-inflammatory activities, which may be a possible mechanism of AM on constipation” (Discussion, line 422-425).
Q6. Which of the three components of Allium mongolicum Regel has the beneficial effect to manage functional constipation
A6. Compared with normal group, treatment with compounds 1-3 significantly increased the active amplitude at 50 µM by 157.4%±20.83, 151.5%±17.05 and 147.2%±15.31 fold of normal isolated intestine. There was no significant difference among these three data. Hence, the three components of AM have similarly beneficial effect to manage functional constipation.
Round 2
Reviewer 2 Report
It is a good document. I think it can be published. thanks